# Exact natural gradient in deep linear networks and application to the nonlinear case

**Alberto Bernacchia**
Department of Engineering
University of Cambridge
Cambridge, UK, CB2 1PZ
ab2347@cam.ac.uk

**Máté Lengyel**
Department of Engineering    Department of Cognitive Science
University of Cambridge       Central European University
Cambridge CB2 1PZ, UK        Budapest H-1051, Hungary
m.lengyel@eng.cam.ac.uk

**Guillaume Hennequin**
Department of Engineering
University of Cambridge
Cambridge, UK, CB2 1PZ
g.hennequin@eng.cam.ac.uk

## Abstract

Stochastic gradient descent (SGD) remains the method of choice for deep learning, despite the limitations arising for ill-behaved objective functions. In cases where it could be estimated, the *natural* gradient has proven very effective at mitigating the catastrophic effects of pathological curvature in the objective function, but little is known theoretically about its convergence properties, and it has yet to find a practical implementation that would scale to very deep and large networks. Here, we derive an exact expression for the natural gradient in deep linear networks, which exhibit pathological curvature similar to the nonlinear case. We provide for the first time an analytical solution for its convergence rate, showing that the loss decreases exponentially to the global minimum in parameter space. Our expression for the natural gradient is surprisingly simple, computationally tractable, and explains why some approximations proposed previously work well in practice. This opens new avenues for approximating the natural gradient in the nonlinear case, and we show in preliminary experiments that our online natural gradient descent outperforms SGD on MNIST autoencoding while sharing its computational simplicity.

## 1   Introduction

Stochastic gradient descent (SGD) is used ubiquitously to train deep neural networks, due to its low computational cost and ease of implementation. However, long narrow valleys, saddle points and plateaus in the objective function dramatically slow down learning and often give the illusory impression of having reached a local minimum [Martens, 2010; Dauphin et al., 2014]. The natural gradient is an appealing alternative to the standard gradient: it accelerates convergence by using curvature information, it represents the steepest descent direction in the space of distributions, and is invariant to reparametrization of the network [Amari, 1998; Le Roux et al., 2008]. However, besides some numerical evidence, the exact convergence rate of natural gradient remains unknown, and its implementation remains prohibitive due to its very expensive numerical computation [Pascanu and Bengio, 2013; Martens, 2014; Ollivier, 2015].

In order to gain theoretical insight into the convergence rate of natural gradient descent, we analyze a deep (multilayer) linear network. While deep linear networks have obviously no practical relevance (they can only perform linear regression and are grossly over-parameterized, see below), their

optimization is non-convex and is plagued with similar pathological curvature effects as their nonlinear counterparts. Critically, the dynamics of learning in linear networks are exactly solvable, making them an ideal case study to understand the essence of the deep learning problem and find efficient solutions [Saxe et al., 2013]. Here, we derive an exact expression for the natural gradient in deep linear networks, from which we garner two major insights. First, we prove that the exact natural gradient leads to exponentially fast convergence towards the minimum achievable loss. This, to our knowledge, is the first case where a functional form for the natural gradient's convergence rate has been obtained for an arbitrarily deep multilayer network, and it confirms the long-standing conjecture that the natural gradient mitigates the problem of pathological curvature [Pascanu and Bengio, 2013; Martens, 2014] (and indeed, annihilates it completely in the linear case). Second, our exact solution reveals that the natural gradient can be computed much more efficiently than previously thought. By definition, the natural gradient is the product of the inverse of the $P \times P$ Fisher information matrix $F$ with the $P$-dimensional gradient vector, where $P$ is the number of network parameters (often in the millions) [Yang and Amari, 1998; Amari et al., 2000; Park et al., 2000]. In contrast, our expression exploits the structure of degeneracies in $F$ and requires computing a similar matrix-vector product but in dimension $N$, the *number of neurons in each layer* (in the tens/hundreds). Although this simple expression does not formally apply to the nonlinear case, we adapt it to nonlinear deep networks and show that it outperforms SGD on the MNIST autoencoder problem.

Our exact expression for the natural gradient suggests retrospective theoretical justifications for several previously proposed modifications of standard gradient descent that empirically improved its convergence. In particular, we revisit previous approximations of the Fisher matrix (in the nonlinear case) based on block-diagonal truncations, and provide a possible explanation for their performance (K-FAC, [Martens and Grosse, 2015; Grosse and Martens, 2016; Ba et al., 2016], see also [Heskes, 2000; Povey et al., 2014; Desjardins et al., 2015]). We show that, even in the simple linear case, the exact inverse Fisher matrix is not block-diagonal and the contributions of the off-diagonal blocks to the natural gradient have the same order of magnitude as the on-diagonal blocks. Therefore, contrary to what has been proposed previously, the off-diagonal blocks cannot in principle be neglected. Instead, our analysis reveals that, when taking the inverse and multiplying by the gradient, the off-diagonal blocks of $F$ contribute the exact same terms as the diagonal blocks. This observation is at the core of the surprisingly efficient yet exact way of computing the natural gradient that we propose here.

Finally, our algebraic expression for the natural gradient exhibits similarities with recent, biologically-inspired backpropagation algorithms. To obtain the natural gradient, we show that the error must back-propagate through the (pseudo-)inverses of the weight matrices, rather than their transposes. Multiplication by the matrix pseudo-inverse emerges automatically in algorithms where both forward and backward weights are free parameters [Lillicrap et al., 2016; Luo et al., 2017].

## 2 Natural gradient in deep networks

We consider the problem of learning an input-output relationship on the basis of observed data samples $\{(x_i, y_i)\}$ (input-output pairs) drawn from an underlying, unknown distribution $p^\star(x, y)$. This is achieved by a deep discriminative model, which, given an input $x$, specifies a conditional density $q_\theta(y|x)$ over possible outputs $y$, parameterized by the output layer of a deep network with a set of parameters $\theta$. Specifically, the input vector $x \in \mathbb{R}^{n_0}$ propagates through a network of $L$ layers according to:

$$x_i = \phi_i \left( W_i \, x_{i-1} + b_i \right) \qquad i = 1, \ldots, L \tag{1}$$

where $x_i \in \mathbb{R}^{n_i}$ is the output of layer $i$ (which then serves as an input to layer $i+1$), $W_i \in \mathbb{R}^{n_i \times n_{i-1}}$ is a weight matrix into layer $i$, $b_i \in \mathbb{R}^{n_i}$ is a vector of bias parameters, $\phi_i$ is a function applied element-wise to its vector argument, and $x_0$ is defined as equal to the input $x$. The set of parameters $\theta$ includes all the elements of the weight matrices and bias vectors of all layers, for a total of $P$ parameters. For ease of notation, in the following we include the bias vector $b_i$ for each layer as an additional row in $W_i$, and augment the activation vector $x_{i-1}$ accordingly with one constant component equal to one. The output of the last layer is $x_L$: it depends on all parameters $\theta$ and determines the conditional density $q_\theta(y|x)$, which we assume here is a Gaussian with a mean determined by $x_L$ and a constant covariance matrix, $\tilde{\Sigma}$. Our theoretical results will be obtained for linear networks ($\phi_i(x) = x$ and $b_i = 0$), but we later return to nonlinear networks in numerical simulations.

The above model specifies a joint distribution of input/output pairs, i.e. $p_\theta(x, y) = q_\theta(y|x) q^\star(x)$ where $q^\star(x) = \int dy\, p^\star(x, y)$ is the marginal distribution of the input and does not depend on the parameters $\theta$. The network is trained via maximum likelihood, i.e. by minimizing the following loss function:

$$\mathcal{L}(\theta) = \langle -\log p_\theta(x, y) \rangle_{p^\star} = \langle -\log q_\theta(y|x) \rangle_{p^\star} + \text{const} \tag{2}$$

where the average is over the true distribution $p^\star(x, y)$. In the following, we will use the shorthand notation $\ell(\theta|x, y) = \log p_\theta(x, y)$. Note that, in this setting, maximum likelihood is equivalent to minimizing the KL divergence $D_{\text{KL}}(p^\star \| p_\theta)$ between the true distribution $p^\star(x, y)$ and the model distribution $p_\theta(x, y)$.

A common way of minimizing $\mathcal{L}(\theta)$ is gradient descent, i.e. parameter updates of the form:

$$\frac{d\theta}{dt} \propto -\frac{\partial \mathcal{L}}{\partial \theta} = \left\langle \frac{\partial \ell(\theta|x, y)}{\partial \theta} \right\rangle_{p^\star} \tag{3}$$

where $t$ denotes time elapsed in the optimization process. Although the theory of natural gradient we develop below applies to this continuous-time formulation [Mandt et al., 2017], numerical experiments are performed by discretizing Eq. 3 and setting a finite learning rate parameter. The dynamics of Eq. 3 are guaranteed to decrease the loss function in continuous time when the expectation over $p^\star$ can be evaluated exactly; in practice, these dynamics are approximated by sampling from $p^\star$ using a batch of training data points, and using a small but finite time step (learning rate) – this is SGD.

The natural gradient corresponds to a modification of Eq. 3, which consists of multiplying the (negative) gradient by the inverse of the Fisher information matrix $F$:

$$\frac{d\theta}{dt} \propto -F(\theta)^{-1} \frac{\partial \mathcal{L}}{\partial \theta} \tag{4}$$

where the Fisher information matrix $F \in \mathbb{R}^{P \times P}$ is defined as

$$F(\theta) = \left\langle \frac{\partial \ell}{\partial \theta} \cdot \frac{\partial \ell}{\partial \theta}^T \right\rangle_{p_\theta} \tag{5}$$

Note that the average is taken over the model distribution $p_\theta(x, y) = q_\theta(y|x)q^\star(x)$, rather than the true distribution $p^\star(x, y)$. Since the Fisher matrix is positive definite, the natural gradient also guarantees decreasing loss in continuous time. The Fisher information matrix quantifies the accuracy with which a set of parameters can be estimated by the observation of data, and the natural gradient thus rescales the standard gradient accordingly. The natural gradient has a number of desirable properties: it corresponds to steepest gradient descent in the space of distributions $p_\theta(x, y)$, it is parameterization-invariant, and affords good generalization performance [Amari, 1998; Le Roux et al., 2008]. Moreover, natural gradient descent can be regarded as a second-order method in the space of parameters (e.g. it reduces to the Gauss-Newton method in some cases [Pascanu and Bengio, 2013; Martens, 2014]).

## 3 Exact natural gradient for deep linear networks and quadratic loss

In this paper, we focus on regression problems where the conditional model distribution $q_\theta(y|x)$ is Gaussian, with a mean equal to the output $x_L$ of the last layer of a deep network and some covariance $\tilde{\Sigma}$. Note that other types of distributions can also be used, e.g. a categorical distribution parameterized by the output of a final softmax layer to address classification problems. Using Eq. 2, the loss function for a Gaussian distribution is equal to the mean squared error weighted by the inverse covariance

$$\mathcal{L} = \frac{1}{2} \left\langle (y - x_L)^T \tilde{\Sigma}^{-1} (y - x_L) \right\rangle_{p^\star} + \text{const} \tag{6}$$

where the loss depends on the parameters of the deep network through the conditional mean $x_L$, and the constant includes all the terms that do not depend on $x_L$ and thus on the network parameters. Using the expression for the loss, we can compute the gradient with respect to the weight matrix into layer $i$, as

$$\frac{\partial \mathcal{L}}{\partial W_i} = -\left\langle e_i\, x_{i-1}^T \right\rangle_{p^\star} \tag{7}$$

where $e_i \in \mathbb{R}^{n_i}$ is the error propagated backward to layer $i$ (see below, Eq. 8), and $x_{i-1}$ is the activation of layer $i-1$ propagated forward (Eq. 1). Note that this expression for the gradient is a matrix of the size of $W_i$. The expression for the backpropagated error is given by

$$e_L = \phi'_L \circ \left[ \tilde{\Sigma}^{-1} \left( y - x_L \right) \right]$$
$$e_i = \phi'_i \circ \left[ W_{i+1}^T e_{i+1} \right] \qquad i = 1, \dots, L-1 \qquad (8)$$

where the symbol $\circ$ denotes the element-wise (Hadamard) vector product, and $\phi'_i$ denotes the scalar derivative of $\phi_i$, evaluated at its argument defined in Eq. 1. The gradient is computationally cheap to evaluate, since a single forward pass is used to compute the activations $x_i$ of all layers, and a single backward pass is used to compute the corresponding errors $e_i$.

It is currently unknown if the natural gradient affords an expression as simple and computationally cheap as those used to evaluate the standard gradient (Eqs. 7-8). Here, we derive such an expression in the case of a deep linear network. We thus take $\phi_i(x) = x$ ($\forall i$), and set the bias vectors to zero without loss of generality if the input has zero mean, $\langle x \rangle_{q^\star} = 0$. Using Eq. 1, the activation of the last layer is therefore equal to

$$x_L = \left( W_L \cdot W_{L-1} \cdots W_2 \cdot W_1 \right) x = W x \qquad (9)$$

where we defined the total weight matrix product $W$ in the last expression, equal to the chain of matrix multiplications along all layers $1, 2, \dots, L$. This expression makes obvious the uselessness of having multiple, successive linear layers, as their combined effect reduces to a single one. However, the dynamics of learning (e.g. by gradient descent) in each layer is highly nonlinear, while being amenable to analytical solutions [Saxe et al., 2013].

In the Supplementary Material, we calculate the Fisher information matrix $F$ for a deep linear network. As expected, the Fisher matrix is singular, due to the aforementioned parameter redundancies, and therefore the model cannot be identified in certain directions in parameter space. In particular, the total number of parameters is $P = \sum_{i=1}^{L} n_i n_{i-1}$, where $n_i \times n_{i-1}$ are the dimensions of matrix $W_i$ in layer $i$, and the total number of parameters is obtained by summing over all layers. However, there are only $n_L n_0$ independent parameters, which are the dimensions of the total product of weight matrices, $W$, in Eq. 9. Thus, the Fisher matrix is of rank $n_L n_0$ at most, and is therefore necessarily singular.

Due to the above singularity, the matrix inversion prescribed by Eq. 4 to obtain the natural gradient must be replaced by a generalized inverse (indeed, this is the appropriate way of dealing with this singularity, and it comes from the interpretation of the Fisher matrix as a metric in the space of distributions [Pascanu and Bengio, 2013]). Note that there exist an infinite number of generalized inverses. Our main result is proving that under the natural gradient, the dynamics of $p_\theta$, and therefore also the dynamics of the loss function, are identical for all possible generalized inverses of the Fisher matrix (Supplementary Material). Moreover, any choice thereof leads to exponentially fast convergence towards the minimum loss. Critically though, all those possible generalized inverses might differ greatly in the simplicity and associated computational cost of the resulting parameter updates. We find that one particular generalized inverse leads to the following, remarkably simple expression:

$$\frac{dW_i}{dt} \propto \frac{1}{L} \left\langle e_i e_i^T \right\rangle_{p_\theta}^{-1} \left\langle e_i x_{i-1}^T \right\rangle_{p^\star} \left\langle x_{i-1} x_{i-1}^T \right\rangle_{p_\theta}^{-1} \qquad (10)$$

This expression is equal to the standard gradient (middle term, cf. Eq. 7), multiplied by the inverse covariance of both the backward error $e_i$ (left) and forward activation $x_{i-1}$ (right). Note that these covariances correspond to averages over the model distribution $p_\theta(x, y)$, and *not* the true distribution $p^\star(x, y)$. When the inverses of those covariances do not exist, it is their Moore-Penrose pseudoinverse that must be used instead (Supplementary Material). As expected for the natural gradient, Eq. 10 is dimensionally consistent (weight updates have the same "units" as the weight matrices themselves), and is covariant for linear transformations.

At first glance, Eq. 10 requires two matrix multiplications and inversions per layer, which make it more costly than standard gradient descent. However, if the expectation over $p^\star$ is approximated by sampling as in SGD, then one only needs to perform two matrix-vector products, and make rank-1 updates of $W_i$, which brings the computational cost down to that of SGD. Finally, one can either (pseudo-)invert the two covariance matrices in Eq. 10 e.g. using an SVD (scales poorly with layer size,

but otherwise cache efficient), or directly estimate their inverses using Sherman-Morrison updates (in which case the complexity scales with both layer size and network depth in the same way as for SGD). We discuss these practical issues further below.

## 4 Analytic expression for convergence rate

In this section, we provide a simplified derivation of the exponential decrease of the loss function under the the natural gradient updates given by Eq. 10, which are based on a particular form of the generalized inverse of $F$. The equation for the natural gradient is given by Eq. 16 below, which corresponds to Eq. 34 in the Supplementary Material. A more general derivation of the exponential decrease of the loss function is given in the Supplementary Material, where we show that the same exponential decay of the loss holds for all possible generalized inverses.

Using Eqs. 1 and 8, the forward activation and backward error in a linear network are given by

$$x_{i-1} = (W_{i-1} \cdots W_1) x \tag{11}$$

$$e_i = (W_L \cdots W_{i+1})^T \tilde{\Sigma}^{-1} (y - x_L) \tag{12}$$

Using Eq. 7, the gradient of the loss function is equal to the averaged outer product of the backward error and the forward activity, namely

$$\frac{\partial \mathcal{L}}{\partial W_i} = - \left\langle e_i \, x_{i-1}^T \right\rangle_{p^\star} = - (W_L \cdots W_{i+1})^T \tilde{\Sigma}^{-1} \left\langle (y - x_L) \, x^T \right\rangle_{p^\star} (W_{i-1} \cdots W_1)^T \tag{13}$$

In order to derive the natural gradient update, we calculate the covariance matrices in Eq. 10. The covariance of the backward error is equal to

$$\left\langle e_i e_i^T \right\rangle_{p_\theta} = (W_L \cdots W_{i+1})^T \tilde{\Sigma}^{-1} \left\langle (y - x_L) (y - x_L)^T \right\rangle_{p_\theta} \tilde{\Sigma}^{-1} (W_L \cdots W_{i+1})$$

$$= (W_L \cdots W_{i+1})^T \tilde{\Sigma}^{-1} (W_L \cdots W_{i+1}) \tag{14}$$

The second line results from averaging over the model distribution $p_\theta(x, y) = q_\theta(y|x) q^\star(x)$: the first average over the conditional distribution $q_\theta(y|x) = \mathcal{N}(y; x_L, \tilde{\Sigma})$ yields the covariance $\tilde{\Sigma}$ itself, and the latter does not depend on the input (making the average over $q^\star(x)$ unnecessary). Similar arguments lead to the covariance of the forward activity:

$$\left\langle x_{i-1} x_{i-1}^T \right\rangle_{p_\theta} = (W_{i-1} \cdots W_1) \left\langle xx^T \right\rangle_{p_\theta} (W_{i-1} \cdots W_1)^T = (W_{i-1} \cdots W_1) \Sigma (W_{i-1} \cdots W_1)^T \tag{15}$$

where $\Sigma = \left\langle xx^T \right\rangle_{q^\star}$ is the covariance of the input (the average is taken over the model distribution $p_\theta$, but $xx^T$ depends on the input distribution $q^\star$ only).

In order to compute the natural gradient of Eq. 10, we need to invert the covariances in Eqs. 14 and 15. However, they may not be invertible, except in special cases, such as when all weight matrices are square and invertible, and when both $\Sigma$ and $\tilde{\Sigma}$ are full rank. We consider this simple case first, and then address the general case of non-square matrices. If we can invert explicitly the relevant covariance matrices, substituting into Eq. 10, along with Eq. 13, yields updates of the form

$$\frac{dW_i}{dt} \propto \frac{1}{L} (W_L \cdots W_{i+1})^{-1} \left\langle (y - x_L) x_0^T \right\rangle_{p^\star} \Sigma^{-1} (W_{i-1} \cdots W_1)^{-1} \tag{16}$$

This equation does not immediately suggest any advantage with respect to standard gradient descent. However, it is revealing to derive the dynamics of the total weight matrix product, $W = W_L \cdots W_1$, which represents the net input-output mapping performed by the network. Using the product rule of differentiation:

$$\frac{dW}{dt} = \sum_{i=1}^{L} (W_L \cdots W_{i+1}) \frac{dW_i}{dt} (W_{i-1} \cdots W_1) \tag{17}$$

Substituting the expression for the update, Eq. 16, and using $x_L = W x_0$ we obtain

$$\frac{dW}{dt} \propto -W + \left\langle yx^T \right\rangle_{p^\star} \Sigma^{-1} \tag{18}$$

Thus, under natural gradient descent in continuous time, the total weight matrix obeys first order dynamics, and therefore converges exponentially fast towards $\langle yx^T \rangle \Sigma^{-1}$, which is indeed the least squares solution to the linear regression problem [Bishop, 2016]. Since the loss is a quadratic function of $W$ (cf. Eq. 6), Eq. 18 also proves that the loss decays exponentially towards zero under natural gradient descent. This result holds provided that the network parameters are not initialized at a saddle point (for example, weights should not be initialized at zero).

When the covariances in Eqs. 14 and 15 cannot be inverted, e.g. when the weight matrices are not square (the network is contracting, expanding, or contains a bottleneck), we show in the Supplementary Material (Eq. 45) that the Moore-Penrose pseudo-inverse must be used instead, inducing similar dynamics for $W$:

$$\frac{dW}{dt} \propto -W + \frac{1}{L} \sum_{i=1}^{L} P_i^a \left\langle yx^T \right\rangle_{p^\star} \Sigma^{-1} P_i^b \tag{19}$$

Here, $P_i^a$ and $P_i^b$ are projection matrices that express the way in which the network architecture constrains the space of solutions that the network is allowed to reach. For example, if the network has a bottleneck, the total matrix $W$ will only be able to attain a low-rank approximation of the optimal solution to the regression problem, $\langle yx_0^T \rangle \Sigma^{-1}$. Note, for example, that $P_i^a = I$ (identity matrix) for a non-expanding network, while $P_i^b = I$ for a non-contracting network.

## 5 Implementation of natural gradient descent and experiments

Similar to SGD, we approximate the average over $p^\star$ in Eq. 7 by using mini-batches of size $M$. For each input mini-batch $x$, we use the forward activations (already calculated in the forward pass to get the gradient information) to estimate $\Lambda_i = \left\langle x_{i-1}x_{i-1}^T \right\rangle_{p_\theta}$. Then, for the same input mini-batch, we also sample $K$ times from the model predictive distribution $q_\theta(y|x)$, use these outputs as targets, and perform the corresponding $K$ backward passes to obtain $KM$ backpropagated error samples used to estimate $\tilde{\Lambda}_i = \left\langle e_i e_i^T \right\rangle_{p_\theta}$. Note that the true outputs of the training set are only used to compute (a stochastic estimate of) the gradient of the loss function, but never used to estimate $\Lambda_i$ nor $\tilde{\Lambda}_i$ (indeed, these are averages over $p_\theta$, not $p^\star$). In practice, we find that $K = 1$ suffices.

Weights are updated according to Eq. 10, discretized using a small time step (learning rate $\alpha$). Inspired by the interpretation of NGD as a second-order method [Martens, 2014], we also incorporate a Levenberg-Marquardt-type damping scheme: at each iteration $k$, we add $\sqrt{\lambda_k}I$ to both covariance matrices $\Lambda_i$ and $\tilde{\Lambda}_i$ prior to inverting them, where $\lambda_k$ is an adaptive damping factor. Note that this is not equivalent to adding $\lambda_k$ to the Fisher matrix. Nevertheless, it does become equivalent to a small SGD step in the limit of large damping factor $\lambda_k$. Therefore, at iteration $k$ we update the synaptic weights in layer $i$ according to

$$\Delta W_i = \frac{\alpha}{L} \left( \tilde{\Lambda}_i + \sqrt{\lambda_k}I \right)^{-1} \left\langle e_i x_{i-1}^T \right\rangle_{p^\star} \left( \Lambda_i + \sqrt{\lambda_k}I \right)^{-1} \tag{20}$$

We update $\lambda_k$ in each iteration to reflect the ratio $\rho_k$ between i) the actual decrease in the loss resulting from the latest damped NG parameter update, and ii) the decrease predicted by a quadratic approximation to the loss[1]. The damping factor is updated as follows:

$$\lambda_{k+1} = \begin{cases} \frac{3\lambda_k}{2} & \text{if } \rho_k < 0.25 \\ \frac{2\lambda_k}{3} & \text{if } \rho_k > 0.75 \end{cases} \tag{21}$$

We experimented with deep networks (linear and nonlinear) trained on regression problems (Fig. 1). First, we trained three linear networks to recover the mappings defined by random networks in their model class. The first network (Fig. 1A) had $L = 16$ layers of the same size $n_i = 20$. The second (Fig. 1B) had $L = 16$ layers, of size $20(\text{input}), 30, 40, \ldots, 100, \ldots, 30, 20$. While these two networks

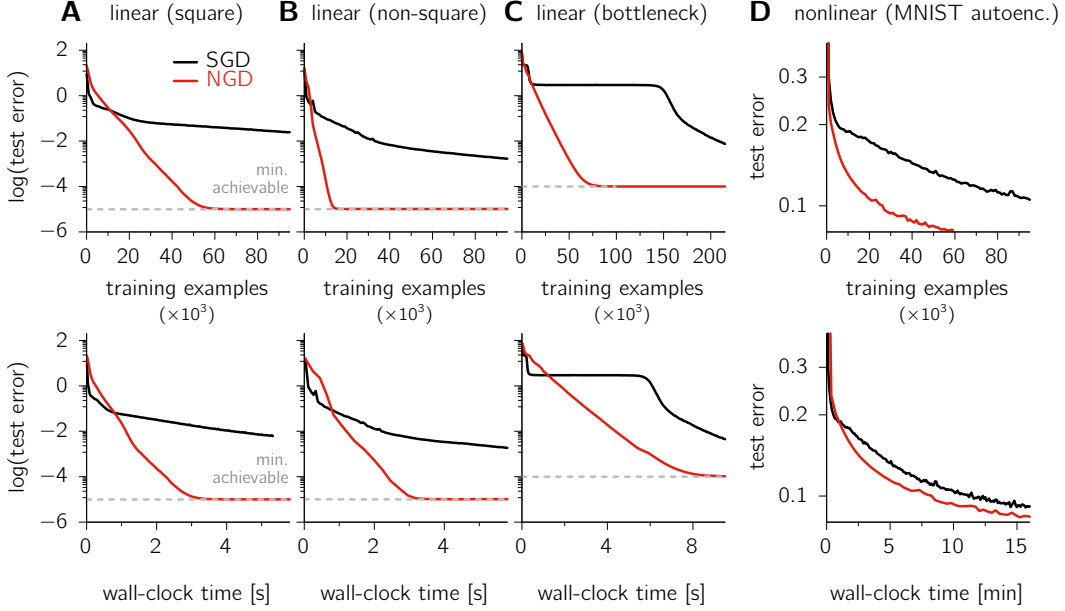

Figure 1: **Natural gradient in deep networks**. **(A–C)** Dynamics of the loss function under optimization with SGD (black) and NGD (red), for three deep linear networks with different architectures (see main text for details). Training time is reported both as number of training examples seen so far (top) and wall-clock time (bottom). Both optimization algorithms start from the same initial network parameters. Dashed gray lines denote the smallest possible loss, determined by the variance of the true underlying conditional density of $y|x$. **(D)** Test error for MNIST autoencoding in a deep nonlinear network (see main text for details); colors are the same as in (A–C). SGD parameters: $M = 20$, learning rate $\alpha$ optimized by grid search (A and B: $\alpha = 0.08$; C: $\alpha = 0.02$; D: $\alpha = 0.04$). NGD parameters: $\alpha = 1$, $M = 1000$.

were over-parameterized, our third network (Fig. 1C) was an under-parameterized bottleneck with steep fan-in and fan-out, with $L = 12$ layers of size $200(\text{input}), 80, 34, 20, 10, 5, 2, 5, \ldots, 80, 200$. For each architecture, we generated a network with random parameters $\theta^\star$ and used it as the reference mapping to be learned. We generated a training set of $10^4$ examples, and a test set of $10^3$ examples, by propagating inputs drawn from a correlated Gaussian distribution $q^\star(x) = \mathcal{N}(x; 0, \Sigma)$ through the network, and sampling outputs from a Gaussian conditional distribution $q_{\theta^\star}(y|x)$ with covariance $\tilde{\Sigma} = 10^{-6}I$. We generated $\Sigma$ to have random (orthogonal) eigenvectors and eigenvalues that decayed exponentially as $e^{-5i/n_0}$.

We compared SGD (with minibatch size $M = 20$, and learning rate optimized via grid search) and online natural gradient (with minibatch size $M = 1000$). For both tasks, SGD made fast initial progress, but slowed down dramatically very soon. In contrast, as predicted by our theory, natural gradient descent caused the test error to decrease exponentially and reach the minimum achievable loss (limited by $\tilde{\Sigma}$) after only a few passes through the training set (Fig. 1A-C, top).

As a preliminary extension to the nonlinear case, we also trained a nonlinear network with eight layers of size $784(\text{input}), 400, 200, 100, 50, 100, \ldots, 784$, to perform autoencoding of the MNIST dataset (Fig. 1D). All layers had $\phi_i(x) = \tanh(x)$, except for the final linear layer. We compared standard SGD (with $M = 20$ and $\alpha$ optimized by grid search) to our proposed natural gradient method (Eq. 10, with adaptive damping and no further modification). We set $\alpha = 1$, $M = 1000$ and $K = 1$. Despite our NGD steps only approximating the true natural gradient, it outperformed SGD in terms of data efficiency (Fig. 1D, top). Owing to the size of the input layer, our implementation of NGD via direct inversion of the relevant covariance matrices outperformed SGD only modestly in wall-clock time (Fig. 1D, bottom). We discuss alternative implementations below.

# 6 Related work

**Diagonal approximations**   As reviewed in Martens [2014], some recent popular methods can be interpreted as diagonal approximations to the Fisher matrix $F$, such as AdaGrad [Duchi et al., 2011], AdaDelta [Zeiler, 2012], and Adam [Kingma and Ba, 2014]. Those methods are computationally cheap, but do not capture pairwise dependencies between parameters. In theory, faster learning could be obtained by leveraging full curvature information, which requires moving away from a purely diagonal approximation of $F$. However, this is computationally intensive for at least two reasons: i) the Fisher matrix is large, often impossible to store, let alone to invert, and ii) even if one could compute $F^{-1}$, the natural gradient would still require $\mathcal{O}(P^2)$ operations (where $P$ is the number of parameters). Much of the recent literature has focused on ways of mitigating this complexity. For example, in cases where it can be stored, $F^{-1}$ can be estimated directly using the Sherman-Morrison lemma [Amari et al., 2000]. When it cannot be stored, one can approximate the natural gradient directly via conjugate gradients, exploiting fast methods for computing $Fv$ products (as in Hessian-free and Gauss-Newton optimization [Martens, 2010; Pascanu and Bengio, 2013; Martens and Grosse, 2015; Vinyals and Povey, 2012]). Often, however, many steps of conjugate gradients must be performed at each training iteration to make good progress on the loss. Here, we have obtained the surprising result that $F^{-1}v$ products can in fact be obtained directly (in linear networks), at almost the same cost as $Fv$.

**Block-diagonal approximations**   In order to obtain an expression for the natural gradient that would be computationally cheap and feasible for practical applications, previous studies suggested a block-diagonal approximation to the inverse Fisher information matrix, in the nonlinear case (K-FAC, [Martens and Grosse, 2015; Grosse and Martens, 2016; Ba et al., 2016], see also [Heskes, 2000; Povey et al., 2014; Desjardins et al., 2015]). In general, there is no formal justification for assuming that the Fisher information matrix (or its inverse) is block diagonal. In our deep linear network model, we show in the Supplementary Material (cf. Eq. 19) that the $(i, j)$-block of the exact Fisher information matrix (corresponding to the weight matrices of layers $i$ and $j$), is equal to

$$F_{ij} = \left\langle x_{i-1} x_{j-1}^T \right\rangle_{p_\theta} \otimes \left\langle e_i e_j^T \right\rangle_{p_\theta} \tag{22}$$

There is no reason to expect that this expression is zero for $i \neq j$, unless the outputs $x_i$ or the errors $e_i$ are uncorrelated across all pairs of layers, and indeed Eq. 19 in the Supplementary Material shows that it is not zero. Nevertheless, if we choose to ignore this fact and set $F_{ij} = 0$ for $i \neq j$, then inverting the Fisher matrix (by inverting separately each diagonal block $F_{ii}$) generates an expression proportional to the exact natural gradient of Eq. 10.

In order to understand this puzzling observation, we recall that the exact Fisher is singular, and we chose a specific form for the generalized inverse $F^g$ in order to derive Eq. 10 (while noting that the dynamics of the loss is the same for all possible inverses). In the Supplementary Material (cf. Eq. 36), we note that the $(i, j)$-block of this specific generalized inverse is equal to

$$(F^g)_{ij} = \frac{1}{L^2} \left\langle x_{j-1} x_{i-1}^T \right\rangle_{p_\theta}^{-1} \otimes \left\langle e_j e_i^T \right\rangle_{p_\theta}^{-1} \tag{23}$$

Thus each block of the inverse Fisher is equal to the inverse of the corresponding block of the (transposed) Fisher matrix (note that we assumed square and invertible blocks). However, the inverse Fisher is not block-diagonal either, thus it remains unclear why the approximation works. The solution to this puzzle is the following. In deriving the natural gradient update for layer $i$, we must multiply an entire row of blocks of the inverse Fisher by the gradient across all layers. Surprisingly, each of these blocks makes exactly the same contribution to the natural gradient (Eq. 37 in the Supplementary Material). Thus, we can get away with computing the single contribution of the diagonal block for each row, and simply multiply that by the number of blocks in the row. This is of course equivalent, though only fortuitously so, to making a block-diagonal approximation of $F$ in the first place. Therefore, somewhat incidentally, a block-diagonal approximation is expected to perform just as well as the full matrix inversion.

**Whitening and biological algorithms**   Our expression for the natural gradient offers post-hoc justifications for some recently proposed modifications of the standard gradient, whereby the forward activation and backward errors are whitened prior to being multiplied to obtain the gradient at each layer [Desjardins et al., 2015; Fujimoto and Ohira, 2018]. In our method, these vectors are

also rescaled, albeit with their inverse covariances instead of the square root thereof (Eq. 10; see also Heskes [2000]; Martens and Grosse [2015]). Notably, this form of rescaling is equivalent to backpropating the error through the (pseudo)-inverses of the weight matrices, rather than their transpose (Eq. 16); interestingly, this strategy also tends to emerge in more biologically plausible algorithms in which both forward and backward weights are free parameters [Lillicrap et al., 2016; Luo et al., 2017].

## 7 Conclusions

We computed the natural gradient exactly for a deep linear network with quadratic loss function. We showed that the natural gradient is not unique in this case, because the Fisher information is singular due to over-parameterization. Surprisingly, we found that the loss function has the same convergence properties for all possible natural gradients, i.e. as obtained by any generalized inverse of the Fisher matrix. Indeed, one of our main results is the first exact solution for the convergence rate of the loss function under natural gradient descent, for a linear multilayer network: exponential decrease towards the minimum loss. This result backs up empirical claims of the natural gradient efficiently optimizing deep networks; in the deep linear case, we find that it solves the problem of pathological curvature entirely [Pascanu and Bengio, 2013; Martens, 2014]. Our results also consolidate deep linear networks as a useful case study for advancing the theory of deep learning. While Saxe et al. [2013] used linear theory to propose new ways of initializing neural networks, we have used it to propose a new, efficient optimization algorithm. We found that natural gradient updates afford an unexpectedly cheap form, with similar computational requirements as plain SGD.

Compared with the size of deep neural networks currently used, our application concerned relatively small networks of at most a few hundreds neurons per layer. Our current implementation based on direct inversion of $\Lambda$ and $\tilde{\Lambda}$ in Eq. 20 may scale poorly (in wall-clock time) as the layer sizes increase. In this case, matrix pseudo-inversion in Eq. 10 could be performed using randomized SVD algorithms [Halko et al., 2011]. Alternatively, direct estimation of those matrix inverses via the Sherman-Morrison (SM) lemma should scale better [Amari et al., 2000], which we have confirmed in preliminary simulations. As SM updates tend to be less cache-efficient than direct inversion (they require many matrix-vector products instead of fewer matrix-matrix products), they may only benefit performance for very large layers. Moreover, more work is needed to incorporate adaptive damping into SM estimation of inverse covariances.

Our analytical results were derived for continuous time optimization dynamics. While we presented numerical evidence showing that a discrete-time implementation of NGD performs well, and it indeed shows the exponential decrease of the loss function predicted by our theory, further work is necessary in order to derive principled methods for discretizing the parameter updates [Martens, 2014].

Our core results relied exclusively on linear activation functions. While we have had some success in training nonlinear networks using Eq. 10 as a drop-in replacement for SGD (Fig. 1D), much remains to be done to make our algorithm effective in general deep learning settings. Improvements could be made to our adaptive damping scheme, for example through asymmetric damping of the covariance matrices $\Lambda_i$ and $\tilde{\Lambda}_i$ in Eq. 20 as proposed by Martens and Grosse [2015]. More generally, deeper links need to be established between our linear NGD theory and systematic methods based on Kronecker factorizations (K-FAC [Martens and Grosse, 2015; Grosse and Martens, 2016; Ba et al., 2016]). A key insight from our analysis is that there exist infinitely many ways of computing the NG in linear deep networks (and probably also in nonlinear networks in which the Fisher matrix has been found to be near-degenerate [Le Roux et al., 2008]). While all of these different methods result in fast learning with identical dynamics for the loss function, their computational complexity may differ greatly. Moreover, there may be more than one computationally tractable method (such as the one we have used here), and in turn, some of these may be more suitable than others for use as a drop-in replacement to SGD in nonlinear networks. We suggest that further analysis of deep linear networks will prove invaluable for deriving efficient new training algorithms.

## Acknowledgments

We thank Richard Turner and James Martens for discussions. This work was supported by Wellcome Trust Seed Award 202111/Z/16/Z (G.H.) and Wellcome Trust Investigator Award 095621/Z/11/Z (A.B.,M.L.).

## Footnotes

[1]Here, the quadratic approximation is implicitly defined as the quadratic function whose minimization by the Newton method would require a step in the direction of $\Delta W_i$, the momentary update taken by our damped NGD step. The predicted decrease in loss under such a quadratic approximation is cheap to compute: if $\Delta W_i$ is the NG update for layer $i$, then the predicted decrease in the loss is given by $\left( -\alpha + \frac{\alpha^2}{2} \right) \sum_i \text{tr} \left( \Delta W_i^T \left\langle e_i x_{i-1}^T \right\rangle_{p^\star} \right)$.

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
