[Supplementary Material]

# Exact natural gradient in deep linear networks and application to the nonlinear case
# Supplementary Material

**Alberto Bernacchia**
Department of Engineering
University of Cambridge
Cambridge, UK, CB2 1PZ
ab2347@cam.ac.uk

**Máté Lengyel**
Department of Engineering    Department of Cognitive Science
University of Cambridge    Central European University
Cambridge CB2 1PZ, UK    Budapest H-1051, Hungary
m.lengyel@eng.cam.ac.uk

**Guillaume Hennequin**
Department of Engineering
University of Cambridge
Cambridge, UK, CB2 1PZ
g.hennequin@eng.cam.ac.uk

## 1 Calculation of the natural gradient

In this section we calculate exactly the natural gradient in a deep linear network with Gaussian noise. We show that the natural gradient is not unique, but the dynamics of the loss function does not depend on its choice: all possible natural gradients lead to exponentially fast minimization of the loss. We show that one specific choice of the natural gradient is equivalent to the rule studied in the main text, Eq.(10) and Eq.(16).

### 1.1 Standard gradient

Here we summarized the main assumptions of the model and calculate the gradient of the loss function. The output $y$ of the deep network is specified by the distribution $q_\theta(y|x)$, conditioned on the input $x$, a Gaussian with mean $x_L$ and covariance $\tilde{\Sigma}$, where $x_L$ is the output of the last layer of the deep network, which depends on the parameters $\theta$ (all synaptic weights in this section). In a deep linear network, $x_L$ is equal to

$$x_L = Wx \tag{1}$$

where $W$ is the total weight matrix product across layers, given by

$$W = W_L \cdots W_1 \tag{2}$$

and $L$ is the total number of layers. The number of neurons in layer $i$ is $n_i$, and the number of components of the input is $n_0$; The matrix $W_i$ has size $n_i \times n_{i-1}$. The log-likelihood of the conditional mean $x_L$, given the observed data $(x, y)$, is equal to

$$\ell(x_L|x, y) = \log q_\theta(y|x) + \text{const} = -\frac{1}{2}(y - x_L)^T \tilde{\Sigma}^{-1}(y - x_L) + \text{const} \tag{3}$$

Note that the log-likelihood depends on the parameters $W$ through the conditional mean $x_L$. Constant terms do not depend on $x_L$ and therefore on $W$. The loss function is defined as minus the log-likelihood (only the relevant term) averaged over the true distribution $p^\star(x, y)$, namely

$$\mathcal{L}(W) = \langle -\ell(x_L|x, y) \rangle_{p^\star} = \frac{1}{2} \left\langle (y - Wx)^T \tilde{\Sigma}^{-1} (y - Wx) \right\rangle_{p^\star} \tag{4}$$

The goal is to minimize this loss function with respect to the parameters $W_1, \ldots, W_L$. For convenience of notation, we define the product of weight matrices *ahead* of a given layer $i$, equal to

$$W_i^a = W_L \cdots W_{i+1} \tag{5}$$

and the product of weight matrices *behind* a given layer $i$, equal to

$$W_i^b = W_{i-1} \cdots W_1 \tag{6}$$

Such that the total weight matrix product is rewritten as $W = W_i^a W_i W_i^b$ for any layer $i = 1, \ldots, L$. Using all definitions above, we calculate the gradient of the log likelihood with respect to weight matrix $W_i$ in layer $i$, which is equal to

$$\frac{\partial \ell}{\partial W_i} = e_i x_{i-1}^T = W_i^{aT} \frac{\partial \ell}{\partial x_L} x^T W_i^{bT} \tag{7}$$

where the backward error and forward activity are equal to

$$e_i = W_i^{aT} \frac{\partial \ell}{\partial x_L} \qquad x_{i-1} = W_i^b x \tag{8}$$

respectively a vector of $n_i$ and $n_{i-1}$ components, and the gradient is a matrix of size $n_i \times n_{i-1}$. The gradient of the log likelihood with respect to the network output, for a Gaussian distribution, is equal to the error weighted by the (inverse) conditional covariance, namely

$$\frac{\partial \ell}{\partial x_L} = \tilde{\Sigma}^{-1} (y - x_L) \tag{9}$$

Standard gradient descent corresponds to updating the weights according to minus the gradient of the loss function, or plus the gradient of the averaged log-likelihood, namely

$$\frac{dW_i}{dt} \propto -\frac{\partial \mathcal{L}}{\partial W_i} = \left\langle \frac{\partial \ell}{\partial W_i} \right\rangle_{p^\star} = \left\langle e_i x_{i-1}^T \right\rangle_{p^\star} \tag{10}$$

## 1.2 Exact Fisher information matrix

The natural gradient is obtained by calculating and inverting the Fisher information matrix, and then multiplying the result by the gradient, as described by Eq.(4). Note that the gradient for layer $i$, Eq.(7), is a matrix of size $n_i \times n_{i-1}$; In order to calculate the Fisher information matrix, we need to express the gradient in vectorized form, where the columns of the matrix are piled up in a single column vector of $n_i n_{i-1}$ components. Using Eq.(7), the vectorized form of the gradient for layer $i$ is equal to

$$\frac{\partial \ell}{\partial \text{Vec}(W_i)} = \text{Vec}\left(W_i^{aT} \frac{\partial \ell}{\partial x_L} x^T W_i^{bT}\right) = \tag{11}$$

$$= \left(W_i^b \otimes W_i^{aT}\right) \text{Vec}\left(\frac{\partial \ell}{\partial x_L} x^T\right) = (W_i^b x) \otimes \left(W_i^{aT} \frac{\partial \ell}{\partial x_L}\right) \tag{12}$$

where we used the definition of the Kronecker product $\otimes$. The Fisher information matrix $\mathbf{F}$ is given by Eq.(5); The outer products of gradients is calculated by noting that the vector of all parameters $\theta$ piles up the elements of all weight matrices, and is written in vectorized form as $\theta = \text{Vec}(W_1, \ldots, W_L)$. Therefore, the Fisher information matrix is equal to

$$\mathbf{F} = \left\langle \frac{\partial \ell}{\partial \text{Vec}(W_1, \ldots, W_L)} \cdot \frac{\partial \ell}{\partial \text{Vec}(W_1, \ldots, W_L)}^T \right\rangle_{p_\theta} \tag{13}$$

where all columns of all matrices are piled up in a single vector with $P = \sum_{i=1}^L n_i n_{i-1}$ components, which represents the total number of parameters, and the Fisher matrix has size $P \times P$. In order to simplify the notation, we consider different blocks of this matrix, each block corresponding to a different pair of weight matrices (a different pair of layers). The $(i, j)$-block of the Fisher matrix $\mathbf{F}$ is given by

$$\mathbf{F}_{ij} = \left\langle \frac{\partial \ell}{\partial \text{Vec}(W_i)} \cdot \frac{\partial \ell}{\partial \text{Vec}(W_j)}^T \right\rangle_{p_\theta} \tag{14}$$

This block has size $n_i n_{i-1} \times n_j n_{j-1}$. Using Eq.(12) and the mixed product property of the Kronecker product, this is equal to

$$\mathbf{F}_{ij} = \left\langle \left( W_i^b x x^T W_j^{bT} \right) \otimes \left( W_i^{aT} \frac{\partial \ell}{\partial x_L} \frac{\partial \ell}{\partial x_L}^T W_j^a \right) \right\rangle_{p_\theta} = \tag{15}$$

$$= \left( W_i^b \otimes W_i^{aT} \right) \left\langle x x^T \otimes \frac{\partial \ell}{\partial x_L} \frac{\partial \ell}{\partial x_L}^T \right\rangle_{p_\theta} \left( W_j^b \otimes W_j^{aT} \right)^T \tag{16}$$

Again, this expression is averaged over the model distribution $p_\theta(x, y) = q_\theta(y|x) q^\star(x)$. Since the conditional distribution $q_\theta(y|x)$ is assumed Gaussian, the two factors inside angular brackets can be averaged separately; The left factor $x x^T$ does not depend on $y$, thus the right factor can be separately averaged over $q_\theta(y|x)$. Once averaged, the right factor gives the inverse conditional covariance (see Eq.(9)), which does not depend on $x$ for a Gaussian distribution, and the left factor can be then averaged separately over $x$. Specifically, we have that

$$\left\langle x x^T \otimes \frac{\partial \ell}{\partial x_L} \frac{\partial \ell}{\partial x_L}^T \right\rangle_{p_\theta} = \left\langle x x^T \otimes \left\langle \frac{\partial \ell}{\partial x_L} \frac{\partial \ell}{\partial x_L}^T \right\rangle_{q_\theta} \right\rangle_{q^\star} = \left\langle x x^T \right\rangle_{q^\star} \otimes \tilde{\Sigma}^{-1} = \Sigma \otimes \tilde{\Sigma}^{-1} \tag{17}$$

where we defined $\Sigma = \left\langle x x^T \right\rangle$ as the covariance of the input (assumed centered - zero mean), and we used the fact that $q_\theta$ is Gaussian. Therefore, the $(i, j)$-block of the Fisher matrix is equal to

$$\mathbf{F}_{ij} = \left( W_i^b \otimes W_i^{aT} \right) \left( \Sigma \otimes \tilde{\Sigma}^{-1} \right) \left( W_j^b \otimes W_j^{aT} \right)^T \tag{18}$$

By the mixed product property of the Kronecker product, this can be rewritten as

$$\mathbf{F}_{ij} = \left( W_i^b \Sigma W_j^{bT} \right) \otimes \left( W_i^{aT} \tilde{\Sigma}^{-1} W_j^a \right) = \left\langle x_{i-1} x_{j-1}^T \right\rangle_{p_\theta} \otimes \left\langle e_i e_j^T \right\rangle_{p_\theta} \tag{19}$$

where we used the definition of backward error and forward activity, Eq.(8). This expression is exact in the linear Gaussian case studied here, and was derived previously as an approximation of the nonlinear case [Heskes, 2000, Povey et al., 2014, Desjardins et al., 2015, Martens and Grosse, 2015, Grosse and Martens, 2016, Ba et al., 2016]).

## 1.3 Generalized inverse Fisher matrix

In order to calculate the natural gradient, the full Fisher matrix needs to be inverted, including all blocks. For convenience of notation, we define the following matrix

$$A = \left( \left( W_1^b \otimes W_1^{aT} \right)^T, \ldots, \left( W_L^b \otimes W_L^{aT} \right)^T \right)^T \tag{20}$$

Note that this matrix has size $P \times n_0 n_L$, and it represents the Jacobian of the function $W = W_L \cdots W_1$. We also define the following matrix, which corresponds to the Fisher information for a neural network with a single layer

$$B = \left( \Sigma \otimes \tilde{\Sigma}^{-1} \right) \tag{21}$$

which is square, positive definite and invertible, by the assumption that input and output are full rank, and has size $n_0 n_L \times n_0 n_L$. Then, putting together all blocks of Eq.(18), the full $P \times P$ Fisher matrix is equal to

$$\mathbf{F} = A B A^T \tag{22}$$

By looking at the size of matrices $A$ and $B$, it is clear that the Fisher matrix has at most rank $n_0 n_L$ and is not invertible. This is not surprising: since the feedforward network is linear, only $n_0 n_L$ parameters are independent, representing the product of the input and output components. However, we consider the generalized inverse of the Fisher matrix, and we ignore all directions in parameters space for which no information can be obtained.

We assume that $A$ has maximum rank (it has independent columns), then it has a left inverse $A^l$, defined by $A^l A = 1$ (identity matrix). This holds typically when all weight matrices are full rank and the network does not have any bottleneck. Then, the generalized inverse of the Fisher matrix is

$$\mathbf{F}^g = A^{lT} B^{-1} A^l \tag{23}$$

Note that the left inverse of $A$ is non-unique, therefore the generalized inverse of the Fisher matrix is non-unique. However, we show below that the loss decays exponentially towards its minimum, regardless of the choice of $A^l$.

## 1.4 All natural gradients imply exponential decay for the loss

We seek to update the weights according to the natural gradient Eq.(4), which is given by the generalized inverse, Eq.(23), multiplied by the gradient across all weight matrices. Note that the vector of all parameters $\theta$, piling up the elements of all weight matrices, is written in vectorized form as $\theta = \text{Vec}(W_1, \ldots, W_L)$, and the update takes a similar vectorized form. Therefore, the natural gradient updates can be written as

$$\text{Vec}\left(\frac{dW_1}{dt}, \ldots, \frac{dW_L}{dt}\right) \propto A^{l^T} B^{-1} A^l \left\langle \frac{\partial \ell}{\partial \text{Vec}(W_1, \ldots, W_L)} \right\rangle_{p^\star} \tag{24}$$

Note that the average of the gradient is taken over the true distribution $p^\star(x, y)$. This expression can be simplified by noting that the vectorized gradient across all layers, using Eqs.(12,20), can be written as

$$\frac{\partial \ell}{\partial \text{Vec}(W_1, \ldots, W_L)} = A \, \text{Vec}\left(\frac{\partial \ell}{\partial x_L} x^T\right) \tag{25}$$

Therefore, using the left inverse $A^l A = 1$, the natural gradient update is equal to

$$\text{Vec}\left(\frac{dW_1}{dt}, \ldots, \frac{dW_L}{dt}\right) \propto A^{l^T} B^{-1} \text{Vec}\left\langle \frac{\partial \ell}{\partial x_L} x^T \right\rangle_{p^\star} \tag{26}$$

Although we eliminated one of the two left inverses in Eq.(24), an explicit expression for the left inverse $A^l$ is still necessary in order to find a simple formula for the natural gradient. We give such an expression below, but first we show that the total weight matrix product converges exponentially to the optimal solution, if and only if it is updated using the natural gradient, for any choice of the left inverse.

The update for the total weight matrix product is given by

$$\frac{dW}{dt} = \sum_i W_i^a \frac{dW_i}{dt} W_i^b \tag{27}$$

Using again the definition of $A$, Eq.(20), we rewrite this update in vectorized form

$$\text{Vec}\left(\frac{dW}{dt}\right) = A^T \text{Vec}\left(\frac{dW_1}{dt}, \ldots, \frac{dW_L}{dt}\right) \tag{28}$$

Substituting Eq.(26) into Eq.(28), and using again the left inverse, $A^T A^{l^T} = \left(A^l A\right)^T = 1$, we find

$$\text{Vec}\left(\frac{dW}{dt}\right) \propto B^{-1} \text{Vec}\left\langle \frac{\partial \ell}{\partial x_L} x^T \right\rangle_{p^\star} \tag{29}$$

Perhaps surprisingly, the update for the total weight matrix product does not depend on the left inverse $A^l$, and thus it does not depend on the specific choice of the generalized inverse Fisher. Using Eqs.(9,21), the update of the total weight matrix product is equal to

$$\frac{dW}{dt} \propto -W + \left\langle yx^T \right\rangle_{p^\star} \Sigma^{-1} \tag{30}$$

Therefore, the exact natural gradient dynamics predict exponential convergence towards $W = \left\langle yx^T \right\rangle \Sigma^{-1}$, which is indeed the optimal solution of the linear regression problem, regardless of which specific generalized inverse is chosen for the Fisher matrix. Furthermore, exponential convergence holds only if weight matrices are updated following a natural gradient. Instead of the natural gradient in Eq.(24), we may update the weight matrices according to an arbitrary matrix $C$ of size $P \times P$, namely

$$\text{Vec}\left(\frac{dW_1}{dt}, \ldots, \frac{dW_L}{dt}\right) = C \left\langle \frac{\partial \ell}{\partial \text{Vec}(W_1, \ldots, W_L)} \right\rangle_{p^\star} \tag{31}$$

Then, using Eqs.(25,28), the update of the total weight matrix product is given by

$$\text{Vec}\left(\frac{dW}{dt}\right) = \left(A^T C A\right) \text{Vec}\left\langle \frac{\partial \ell}{\partial x_L} x^T \right\rangle_{p^\star} \tag{32}$$

In order to achieve exponential dynamics for any output $y$ (thus for any matrix $\left\langle yx^T \right\rangle$), we must have $A^T C A = B^{-1}$. There are infinitely possible matrices $C$ satisfying this expression, and they can be written as $C = A^{l^T} B^{-1} A^l$, for all possible left inverses $A^l$, but this is indeed a generalized inverse of the Fisher matrix, Eq.(23).

## 1.5 A simple natural gradient for square matrices

We would like to obtain a simple expression for the natural gradient across the single layer matrices $W_1, \ldots, W_L$, using Eq.(26). We note that, if all weight matrices are square and invertible, one possible left inverse is given by (cf Eq. 20)

$$A^l = \frac{1}{L}\left(\left(W_1^{b^{-1}} \otimes W_1^{a^{T-1}}\right), \ldots, \left(W_L^{b^{-1}} \otimes W_L^{a^{T-1}}\right)\right) \tag{33}$$

We show that this expression implies exactly the update rule studied in the main text (Eq.10 and Eq.16 of the main text). Substituting Eqs.(9,21,33) into Eq.(26), we have

$$\text{Vec}\left(\frac{dW_i}{dt}\right) = \frac{1}{L}\text{Vec}\left(W_i^{a-1}\left\langle(y - x_L)\,x^T\right\rangle_{p^\star}\Sigma^{-1}W_i^{b-1}\right) \tag{34}$$

This expression is equal to Eq.(16) of the main text, thus demonstrating that indeed represents an instance of the natural gradient.

As mentioned in the main text, this exact expression is proportional to the block-diagonal approximation of the Fisher matrix, even though the Fisher matrix is not block-diagonal. In order to see this, we invert the diagonal blocks of Eq.(19) and multiply by the gradient Eqs.(12,9)

$$(\mathbf{F}_{ii})^{-1}\left\langle\frac{\partial\ell}{\partial\text{Vec}\,(W_i)}\right\rangle_{p^\star} = \text{Vec}\left(W_i^{a-1}\left\langle(y - x_L)\,x^T\right\rangle_{p^\star}\Sigma^{-1}W_i^{b-1}\right) \tag{35}$$

which is equal to Eq.(34), besides a factor $L^{-1}$. In order to understand this puzzling observation, we look at the $(i, j)$-block of the generalized inverse Fisher matrix, using Eqs.(23), and substituting the left inverse of Eq.(33), that gives

$$(\mathbf{F}^g)_{ij} = \frac{1}{L^2}\left(W_i^{b^{T-1}}\Sigma^{-1}W_j^{b-1}\right) \otimes \left(W_i^{a-1}\tilde{\Sigma}W_j^{a^{T-1}}\right) \tag{36}$$

Comparing this expression with the block-wise Fisher matrix, Eq.(19), we find that the $(i, j)$-block of the inverse Fisher is equal to the inverse of the (transposed) $(j, i)$-block of the Fisher matrix, besides a factor $L^{-2}$. Therefore, the Fisher matrix can be inverted by inverting single blocks individually. Furthermore, each block of the inverse contributes the same amount to the natural gradient, as shown by multiplying the last expression by the gradient of layer $j$ (Eqs. 12, 9), namely

$$(\mathbf{F}^g)_{ij}\left\langle\frac{\partial\ell}{\partial\text{Vec}\,(W_j)}\right\rangle = \frac{1}{L^2}\text{Vec}\left(W_i^{a-1}\left\langle(y - x_L)\,x^T\right\rangle_{p^\star}\Sigma^{-1}W_i^{b-1}\right) \tag{37}$$

This expression does not depend on $j$, therefore each block of the inverse Fisher, across columns, contributes equally to the natural gradient.

## 1.6 A simple update for rectangular matrices

In this section we show that, even if the weight matrices are not square and not invertible, Eq.(10) of the main text approximately results in exponential decay towards the minimum loss, provided that the Moore-Penrose pseudo-inverse is used when inverting covariances (Eqs. 14,15 in the main text).

In order to simplify the notation, we define the following matrices $E_i$, $X_i$

$$\left\langle e_i e_i^T\right\rangle_{p^\star} = E_i E_i^T \qquad\qquad E_i = (W_L \cdots W_{i+1})^T \tilde{\Sigma}^{-1/2} \tag{38}$$

$$\left\langle x_{i-1} x_{i-1}^T\right\rangle_{p^\star} = X_i X_i^T \qquad\qquad X_i = (W_{i-1} \cdots W_1)\Sigma^{1/2} \tag{39}$$

Using the properties of the pseudo-inverse of a product, we can compute explicitly the pseudo-inverse (denoted by the superscript $+$) of both covariance matrices, equal to

$$\left\langle e_i e_i^T\right\rangle_{p^\star}^+ = E_i^{+T} E_i^+ \qquad\qquad \left\langle x_{i-1} x_{i-1}^T\right\rangle_{p^\star}^+ = X_i^{+T} X_i^+ \tag{40}$$

Using the definitions of $E_i$ and $X_i$, we note that the gradient Eq.(13) of the main text can be rewritten as

$$\frac{\partial\ell}{\partial W_i} = E_i\,\tilde{\Sigma}^{-1/2}\,(y - x_L)\,x^T\Sigma^{-1/2}\,X_i^T \tag{41}$$

We calculate the natural gradient update, Eq.(10) of the main text, with the pseudo-inverse in place of the inverse. Using Eqs.(40,41), that is equal to

$$\frac{dW_i}{dt} \propto \frac{1}{L} E_i^{+T} E_i^+ E_i \, \tilde{\Sigma}^{-1/2} \left\langle (y - x_L) \, x^T \right\rangle_{p^\star} \Sigma^{-1/2} \, X_i^T X_i^{+T} X_i^+ \tag{42}$$

Using the properties of the pseudo-inverse, we have that $E_i^{+T} E_i^+ E_i = E_i^{+T}$ and $X_i^T X_i^{+T} X_i^+ = X_i^+$. Furthermore, we use $x_L = W x$, where we rewrite $W = \tilde{\Sigma}^{1/2} E_i^T W_i X_i \Sigma^{-1/2}$. Then the previous expression is rewritten as

$$\frac{dW_i}{dt} \propto \frac{1}{L} E_i^{+T} \left( \tilde{\Sigma}^{-1/2} \left\langle y x^T \right\rangle_{p^\star} \Sigma^{-1/2} - E_i^T W_i X_i \right) X_i^+ \tag{43}$$

We would like to obtain an update for the total weight matrix product, similar to Eq.(18). Substituting Eq.(43) into the product rule Eq.(27), and using again the definitions of $E$ and $X$, the update for the total weight matrix product is equal to

$$\frac{dW}{dt} \propto \frac{1}{L} \sum_{i=1}^{L} \tilde{\Sigma}^{1/2} E_i^T E_i^{+T} \left( \tilde{\Sigma}^{-1/2} \left\langle y x^T \right\rangle_{p^\star} \Sigma^{-1/2} - E_i^T W_i X_i \right) X_i^+ X_i \Sigma^{-1/2} \tag{44}$$

Using again the properties of the pseudo-inverse, $E_i^T E_i^{+T} E_i^T = E_i^T$, and $X_i X_i^+ X_i = X_i$, we finally obtain

$$\frac{dW}{dt} \propto -W + \frac{1}{L} \sum_{i=1}^{L} P_i^a \left\langle y x^T \right\rangle_{p^\star} \Sigma^{-1} P_i^b \tag{45}$$

where $P_i^a$ and $P_i^b$ are projection matrices, defined by $P_i^a = \tilde{\Sigma}^{1/2} \left( E_i^+ E_i \right)^T \tilde{\Sigma}^{-1/2}$ and $P_i^b = \Sigma^{1/2} X_i^+ X_i \Sigma^{-1/2}$. Therefore, the total product matrix converges exponentially to the optimal solution (cf Eq.(18) in the main text). The projection operators $P_i^a$ and $P_i^b$ depend on the weight matrices; the optimal solution is projected into a subspace which depends on the specific form of the deep network (e.g., whether is contracting, expanding, or has a bottleneck). Note that this result was obtained assuming that covariances in Eq.(10) of the main text are inverted using the Moore-Penrose pseudoinverse, and may not hold when using a different kind of inverse.