[Reviews · NeurIPS 2018]

Reviewer 1



Update after rebuttal: I continue to think this paper presents a significant result, which may spawn a variety of future follow-ups. Deep linear networks with the L2 loss are undoubtedly a simple model class, but they are nonconvex and exhibit several of the pathologies in nonlinear networks. It is great to have a clear, exact analysis of a way of solving this issue. More generally, I think the results in this paper are an essential prerequisite for tackling more complex versions of the problem. If we don't understand how NG works in this simple linear case, why should we expect to jump straight to a more complex situation? Pedagogically I find this paper very useful in understanding how NG is improving the situation (at least from an optimization perspective). The paper should explicitly discuss and expand on the connection between the proposed algorithm and K-FAC. The algorithm proposed in this paper appears to be identical, suggesting that K-FAC is exactly correct for the deep linear case. K-FAC has been applied in deep nonlinear settings, at large scale, and this may provide an additional argument for its usefulness. ________________ Summary: This paper derives an exact, efficient expression for the natural gradient for the specific case of deep linear networks. The main result is that the natural gradient completely removes pathological curvature introduced by depth, yielding exponential convergence in the total weights (as though it were a shallow network). The paper traces connections to a variety of previous methods to approximate the Fisher information matrix, and shows a preliminary application of the method to nonlinear networks (for which it is no longer exact), where it appears to speed up convergence. Major comments: This paper presents an elegant analysis of learning dynamics under the natural gradient. Even though the results are obtained for deep linear networks, they are decisive for this case and suggest strongly that future work in this direction could bring principled benefits for the nonlinear case (as shown at small scale in the nonlinear auto encoder experiment). The analysis provides solid intuitions for prior work on approximating second order methods, including an interesting observation on the structure of the Hessian: it is far from block diagonal, a common assumption in prior work. Yet off diagonal blocks are repeats of diagonal blocks, yielding similar results. Regarding the exponential convergence result, it seems like this cannot be fully general. If the initial weights are all zero such that the system begins at a degenerate saddle point, it is hard to see how rescaling the updates can correct this. The projection matrices in Eqn 19 are said to depend only on the network architecture, but it seems they might also depend on the parameter initialization: networks initialized with zero weights might effectively be dropping rank, which cannot be recovered. It seems like the statement should be something like, provided one does not initialize at a saddle point, convergence is exponential (and random initialization will start at a saddle point with probability zero).

Reviewer 2



Update: I have considered the author response. I maintain my assessment. ========================================================== This paper proposed a convergence analysis on natural gradients. The authors show that the natural gradient updates converge exponentially fast for the deep linear networks with no activation functions and no bias terms. To make the updates faster, the authors proposed some approximation techniques as well as an online implementation. The authors also analyzed the Fisher information structure for the deep linear networks and argue that the block-diagonal style approximations [2, 3, 4, 5, 6] to the inverse Fisher information matrix may ignore the effects of the off-diagonal blocks during the updates. Meanwhile, through analyzing the deep linear networks, the authors explained that the off-diagonal terms contribute the same as the diagonal terms in the updates, and hence explain why block-diagonal approximations may work well in practice. The problem addressed by this paper is quite interesting. Analyzing the convergence rates of natural gradient methods for general deep neural networks is an open problem, to best of my knowledge. Also, the analysis on the block-diagonal approximation style methods is interesting since these methods are widely applied by the machine learning community. However, I am not sure if the convergence rate analysis of the deep linear networks is useful or not since these networks may not be useful in practice while the case of general deep neural networks may be quite different with this special case. Strengths of the paper: 1. In my opinion, the research direction of studying the convergence rates of natural gradient updates is interesting and useful, since natural gradients are widely used. 2. Using the deep linear networks to analyze the off-diagonal blocks' contributions on natural gradient updates is a very good idea. This analysis helps us understand why block-diagonal style approximations could work. Weakness: 1. Though the analysis on deep linear networks show us some insights on the convergences rates of natural gradient updates, the case might be very different for general deep neural networks since they contain bias terms and non-linear activation functions, which might make things much more complicated. 2. For the deep linear networks with layer sizes n_0, ...., n_L, only the case when n_0 * n_L > \sum\limits_{l=0}^{L-1} n_l * n_{l + 1} is useful (otherwise, it is better just to apply a single matrix W, but not a deep linear network). However, for both experiments in Figure 1(A) and Figure 1(B), \sum\limits_{l=0}^{L-1} n_l * n_{l + 1} is much larger than n_0 * n_L. It is better to show more results on the more interesting cases, such as networks with bottleneck layers (e.g. auto-encoders in Figure 1(C)). 3. For the MNIST experiment in Figure 1(C), perhaps some more comparisons are necessary. First, only comparing to SGD is not sufficient. Other first-order methods (e.g. Adams and RMSProp) or some second-order methods (e.g. Hessian-Free optimization [1]) are good potential baseline methods. Second, auto-encoding on MNIST is a well-studied task. It would be great if the authors could compare the results with the other work on this task. Third, the authors tried to compare the efficiency of their method against SGD through number of iterations, but single iteration of their method may take longer than that of SGD. It will be better if the comparison on time is also presented. Minor comment: Probably Section 7 is too long. I suggest maybe split it to a section of related work and a section of conclusion is better. References: [1] Martens J. Deep learning via Hessian-free optimization. In ICML 2010 Jun 21 (Vol. 27, pp. 735-742). [2] Heskes, T. (2000). On natural learning and pruning in multilayered perceptrons. Neural Computation, 12(4):881–901. [3] Povey, D., Zhang, X., and Khudanpur, S. (2014). Parallel training of dnns with natural gradient and parameter averaging. arXiv preprint arXiv:1410.7455. [4] Desjardins, G., Simonyan, K., Pascanu, R., et al. (2015). Natural neural networks. In Advances in Neural Information Processing Systems, pages 2071–2079. [5] Martens, J. and Grosse, R. (2015). Optimizing neural networks with kronecker-factored approximate curvature. In International conference on machine learning, pages 2408–2417. [6] Grosse, R. and Martens, J. (2016). A kronecker-factored approximate fisher matrix for convolution layers. In International Conference on Machine Learning, pages 573–582.

Reviewer 3



Update: After reading the rebuttal, I am updating my rating to 6. ------------- This paper studies natural gradient descent for deep linear networks. It derives the expression of the gradient for quadratic loss and shows that natural gradient converges exponentially to the least-squares solution. The authors show that the Fisher information matrix is singular for deep linear networks and one has to resort to a generalized inverse of this matrix to compute the natural gradient. The paper reports an online implementation with successive rank-1 updates for an auto-encoder on MNIST. The authors make an interesting observation that a block-diagonal approximation of the Fisher information matrix is sufficient for the linear case. This paper is very well-written with a good overview of relevant literature. The focus on the linear case seems pedagogical and the paper would improve immensely if the authors provide experimental results on larger networks and harder datasets. I have reservations about (i) using the expressions for Fisher information matrix for the quadratic loss for say, a large-scale supervised learning problem with cross-entropy loss, and (ii) the computational overhead of the matrix-vector products in (10) and the expectation over the distribution p_theta. The speed-boost for the sanity-check-example in Fig. 1a, 1b is not seen in the auto-encoder in Fig. 1c. In particular, the X-axis in Fig. 1 should be wall-clock time instead of the number of iterations to make the comparison with SGD fair. The authors use K=10 forward passes for each sample in the mini-batch to estimate Lambda_i in (20); using the natural gradient is computationally more expensive than SGD. Why does such a small value of K seem sufficient to estimate (10) accurately? The memory complexity of the natural gradient updates (even with block-diagonal approximation) should be high. For instance, for the auto-encoder on lines 241-247, the expression in (19) incurs a large memory overhead which makes the applicability of this approach to larger problems questionable.